# Adaptation to Social-Linguistic Associations in Audio-Visual Speech

**DOI:** 10.3390/brainsci12070845

**Published:** 2022-06-28

**Authors:** Molly Babel

**Affiliations:** Department of Linguistics, University of British Columbia, Vancouver, BC V6T 1Z4, Canada; molly.babel@ubc.ca

**Keywords:** perceptual adaptation, linguistic expectations, social stereotypes, speech in noise, intelligibility

## Abstract

Listeners entertain hypotheses about how social characteristics affect a speaker’s pronunciation. While some of these hypotheses may be representative of a demographic, thus facilitating spoken language processing, others may be erroneous stereotypes that impede comprehension. As a case in point, listeners’ stereotypes of language and ethnicity pairings in varieties of North American English can improve intelligibility and comprehension, or hinder these processes. Using audio-visual speech this study examines how listeners adapt to speech in noise from four speakers who are representative of selected accent-ethnicity associations in the local speech community: an Asian English-L1 speaker, a white English-L1 speaker, an Asian English-L2 speaker, and a white English-L2 speaker. The results suggest congruent accent-ethnicity associations facilitate adaptation, and that the mainstream local accent is associated with a more diverse speech community.

## 1. Introduction

Listeners’ experiences in the linguistic world contribute to the formation and reinforcement of associations between language, people, and social structures. Listeners learn that, for example, females being, on average, smaller in stature, have smaller vocal tracts than men, and thus generally have higher frequency boundaries between, for example, vowels [1] and sibilant fricatives [2]. Such expectations about the relationship between talker size and phonetic realizations arguably assist in processing spoken language more efficiently and adeptly. While phonetic associations related to gender or sex are at least partially rooted in physiological differences [3]—as opposed to being wholly culturally-specific learned patterns, see Johnson [4]—between women and men, listeners also connect pronunciation patterns with completely arbitrary social groups. Drawing upon learned associations, listeners can categorize a speaker on a number of different social identities based on speech samples (e.g., ethnicity [5]) and use inferred social characteristics to guide the categorization of spoken language (e.g., [6]). For overviews of the evidence in support of listeners’ vast sociophonetic knowledge space, see Drager [7] and Hay and Drager [8]. Supported by decades of empirical evidence that listeners jointly track social and linguistic information (see overviews in [9,10]), Kleinschmidt et al. [11] build upon their Bayesian ideal adaptor framework for phoneme identification [12] and present a computational model of how listeners can leverage probabilistic co-patterning of socio-indexical characteristics and linguistic features. This join tracking models how listeners are able to infer both linguistic judgments (e.g., given what is known about this talker socially, was that a /p/ or a /b/?) and social-indexical judgments (e.g., given what is known about the identified linguistic category, what dialect region is that talker from?).

The associations and joint probabilities listeners possess may (typically) originate in veridical experiences, but these expectations and the sociolinguistic knowledge listeners carry can warp their perception of the speech stream. Listeners’ ultimate percepts or decisions about what they heard of a given utterance are influenced by what they *expect* a talker from a particular social category to produce. For example, given acoustically identical perceptual stimuli, New Zealand listeners perceive speakers who seem younger as having a more complete near/square merger, consistent with younger speakers being probabilistically more likely to have merged the sounds Hay et al. [6]. Niedzielski [13] found that listeners from Michigan, USA assumed that an apparent speaker from Ontario, Canada had a different accent from their own (despite this lack of difference) and categorized vowels accordingly. Niedzielski also showed that these Michigan listeners perceived their own accent as patterning more with a mainstream American one, indicating a disconnect between actual and perceived pronunciation in their speech community. This indicates that listener expectations about accents and speech patterns, including their own, affect their perceptual or recognition space.

The current research is focused on listener associations between accent and ethnicity. Associations between accent and ethnicity in English-speaking countries with histories as colonizers present a particular challenge, as the associations are frequently shown to be fallible. The fallibility of accent and ethnicity associations is, of course, not a problem that is unique to English-speaking countries. For example, despite multicultural and diverse non-white demographics in the United States, to be considered maximally “American”, one must be white [14]. This association is implicated in speech studies that tap expectations or stereotypes about who is expected to speak “unaccented” English. For example, an influential set of studies paired photos of a white face and a East Asian face with voices representing native and non-native accents [15,16]. Recent scholarship makes a convincing case for the abandonment of the vague label *native speaker* [17]. The term is used in this manuscript as a short-hand for a perceptibly mainstream accent for a local speech community. This phrasing of “perceptibly mainstream” is intended to signal that what is crucial within the current work is that a talker’s accent is *perceived* as being a member of a particular category or speech community, in spite of an individual having, for example, multiple native languages, as is typical in the local speech community. Mainstream language use, however, often masks this multilingual upbringing. In Kang and Rubin’s work, when the voices were paired with the East Asian face, they were perceived as more accented and were associated with lower accuracy on a cloze task. Kang and Rubin call this outcome reverse linguistic stereotyping, which results in evaluations of low social status that negatively affect speech comprehension. Kang and Rubin’s theory hinges on the listener-valued social prestige, riding on some aspect of volition. However, experience and stereotypes may affect speech processing instead of or in addition to a listener’s inclinations. McGowan [18] pursued an exemplar-theoretic explanation of accent and ethnicity associations that is based on experience and not a listener’s willingness to comprehend. In support of his approach, McGowan found that Mandarin-accented English was more intelligible when paired with an East Asian face than with a white face [18]. This facilitating effect in comprehending L2-accented speech is consistent with expectations about the phonetic patterns associated with a given social group. Using speech from a larger set speakers of Canadian English, Babel and Russell [19] demonstrated a similar effect in a speech in noise task that compared audio-only trials with ones pairing audio with white Canadian or Chinese Canadian faces. They found lower accuracy in the transcription of Chinese-Canadian speech only in combination with Chinese-Canadian faces. This effect was greater for listeners who reported spending more time with Chinese Canadians, suggesting that the findings may not be about negative social associations, but instead involve erroneous ethnicity/accent expectations. Similarly, in another English-speaking context, Gnevsheva [20] found that New Zealand English listeners rated a white German-L1/English-L2 speaker as least-accented when presented with a video, middlingly-accented in an audio-only condition, and most-accented in an audio-visual condition. This contrasts with an ethnically Korean Korean-L1/English-L2 speaker being rated as consistently highly accented in all three conditions. Gnevsheva reasons that while listeners expect an ethnically Korean individual to speak English with a non-native accent, their expectations of a white talker are that they will exhibit a native English accent. The mismatch between this expected native accent and reality in an audio-video condition prompts an increase in perceived accentedness.

Gnevsheva [20] examined perceived accentendness and not intelligibility. This distinction is both theoretically and empirically important. Through a series of experiments, Zheng and Samuel [21] demonstrate the changes in perceived accentedness are better characterized as a change in “interpretation” and not “perception” of the speech. These changes in interpretation may be more malleable than changes in perception, as they may tap into stereotypes more directly. Indeed, presenting listeners with native Dutch passages, Hanulíková [22] found that co-presenting the native speech with a photo of an ethnically Moroccan face did not change intelligibility and only increased ratings of accentedness in adverse listening conditions.

Not all accents are equivalent in their ability to elicit ethnicity and accent associations in speech processing. This may be because particular accents are simply more intelligible (e.g., the signal quality of a mainstream accent may be more robust than an accent with which one has less experience) or because particular accents are socially associated with a more diverse group of talkers (e.g., mainstream accents versus rural regional accents). Evidence for the latter interpretation comes from infants. Infants raised in highly multilingual communities develop ethnicity and language associations from an early age. May et al. [23] demonstrated that 11 month old English-acquiring infants in Vancouver, British Columbia associate Cantonese more strongly with Asian faces than white faces. Eleven month old English-acquiring white infants’ looking times at Asian versus white (static) faces are equivalent when infants are presented with English, demonstrating that infants consider both of the faces equally likely to produce (natively-accented) English. On Cantonese language trials, however, infants looked longer at the Asian faces, suggesting an expectation that the Asian face would be more likely to speak Cantonese. Through a series of experiments, May and colleagues suggest that their results are not simply due to an association of an unfamiliar language with a less familiar face.

Using three groups of listeners—teens, young adults, and elderly adults—Hanulíková [24] assessed the effect of face primes on the intelligibility and perceived accentedness of Standard German, Korean-accented German, and Palatinate German. Perceived accentedness was significantly higher for all speech samples for the elderly adult group when accompanied by an Asian face. Only the standard accent was perceived as more accented in the presence of an Asian face for the teen and younger adult age groups. There was some evidence that listeners found the Korean-accented German more intelligible when co-presented with the Asian face and that the Palatinate German accent was more intelligible when accompanied by a white face. Hanulíková [24] found no effect of intelligibility on the standard German accents. These results also align with there being more diverse associations with mainstream accents compared to regional or non-native accents. Such results reiterate that listeners’ experiences shape and guide their social expectations (see also [19]). In Montréal, Québec, Canada, where bi/multi-lingualism is an embedded aspect of the local culture, listeners appear to be more impervious to the effects of white and South Asian face primes paired with American English, British English, and Indian English voices compared to listeners from Gainesville, Florida, USA, where bi/multi-lingualism is a less valued asset [25].

Whether listeners specifically tailor their expectations to a particular accent—that is, adjusting the anticipated phonetic distributions to align with the pronunciation patterns of particular non-native accent (e.g., Mandarin-accented English)—or engage a more global relaxation mechanism is a matter of debate. Like the targeted adaptation to Mandarin-accented English when presented with an image of an Asian talker found in McGowan [18], Vaughn [26] found that giving listeners information about the identity of an upcoming L1-Spanish/L2-English talker improved transcription accuracy. These results suggest a targeted adaptation mechanism that improved American English listeners’ ability to parse Mandarin- and Spanish-accented English in those respective studies. Melguy and Johnson [27] find evidence that supports a more global adaptation mechanism, showing that listeners who believe the talker exhibits any non-native accent show higher transcription accuracy.

The majority of the literature exploring ethnicity and accent or language associations relies on static photos. The reason for this is likely to allow for more convincing applications of matched-guise techniques in the experimental design. The use of static photos, however, removes a layer of ecological validity, as voices are most often accompanied by moving faces, not static ones. Those moving faces are an important source of phonetic information. Generally, audio-visual speech receives a boost in performance compared to audio-only speech (e.g., [28]). This audio-visual benefit, however, has been shown to be larger for natively accented talkers [29]. Yi and colleagues tested listeners using native and Korean-accented English in audio-only and audio-visual conditions. A greater audio-visual boost was found for the native English speakers. For the Korean-accented speakers, listeners’ performance was predicted by the strength of an association between the categories “Asian” and “foreign”. They conclude less experience with Korean faces inhibits listener ability to exploit the facial movements that are known to aid alignment and boost intelligibility.

The summarized literature suggests that listeners use experiences and stereotypes to buffer expectations that help and hinder the processing of accents. The quality of the evidence is mixed, however, with some finding support for expectations exerting influence in both intelligibility and accentedness (e.g., [19]), only accentedness (e.g., [22]), or a mixed bag (e.g., [24]) when intelligibility and accentedness are investigated in tandem. Whether adaptation to accent and ethnicity associations is targeted or global is also mixed [26,27]; recent evidence in support of both targeted and global adaptation mechanisms at work in lexically-guided perceptual adaptation is presented in Babel et al. [30]. The conflicting results within this body of literature may be expected due to the uniqueness of the subject population—rarely are the social and linguistic experiences of the listener population described at length—and the specific social associations and demographic facts of a speech community. As a case in point, Babel and Russell [19] recruited talkers from a particular suburb with a historic and well-established Cantonese-speaking population, and also informed listeners that the talkers were from this particular suburb. We were leveraging locally-held social associations. Regardless, the conflicting results may also be due to spurious findings in either direction—that is, either in support of the role of social expectations in speech perception or against such a mechanism. These considerations warrant an analysis strategy that offers nuance to interpretation, a focus on effect size, and a side-lining of null-hypothesis significance testing. Bayesian data analysis satisfies these desiderata and is deployed for the current set of research questions.

Those research questions are focused on how listeners adapt to accent and ethnicity associations in naturally-produced audio-visual speech. While varied, the literature generally suggests that listeners should be better at adapting to accent and ethnicity associations that match local stereotypes. We test this with a speech in noise sentence transcription task using naturally produced audio-visual stimuli with speech embedded in −5 dB signal-to-noise ratio (SNR) pink noise. The speech samples come from four talkers who vary in terms of self-identified ethnicity—white and Asian—and whether they speak English as a first or second language. Comparing high predictability training sentences and low predictability test sentences, we expect listeners to adapt more easily to the talkers who match accent and ethnicity stereotypes. Local for the current study is the same urban area as Babel and Russell [19] and May et al. [23], which means that we expect to see a reduction in transcription accuracy between training and test trials for the Asian English-L1 and the white English-L2 speakers due to assumptions that ethnically Asian individuals should be non-native English speakers and ethnically white individuals should be native speakers of English. Being trained on an accent and ethnicity pairing counter to local stereotypes is predicted to make adaptation more difficult due to a mismatch between predicted and perceived signals. The white English-L1 and the Asian English-L2 talkers conform to local stereotypes, and we predict that listeners will adapt more to these talkers, showing generalization from the high predictability training sentences to the low predictability test set.

## 2. Methodology

### 2.1. Materials: Audio-Visual Stimuli

Four female talkers in their twenties were recorded reading high and low predictability sentences from Bradlow and Alexander [31]. Example sentences are provide in Table 1. The full sentence list is available at an OSF repository (accessed on 24 May 2022). The talkers included two first language speakers of Canadian English and two second language English speakers. For both the L1 and L2 pairs, one talker was Asian and the other was white. The Asian English-L2 talker was a native speaker of Mandarin and the white one was a native speaker of Spanish; these speakers were chosen out of convenience. The social demographic labels applied to these speakers—female and either Asian or white—are self-identified categories for each talker.

Audio recordings were digitized at 44.1 kHz using a Sennheiser MKH-416 shotgun microphone connected to a USB Pre-2 amplifier and a PC. Video recordings were made using Panasonic HC-V700M high definition video camera, which also recorded audio. The video recordings included the talkers from the neck up against a white background. The high quality audio recordings were RMS-amplitude normalized and embedded in pink noise at a −5 dB SNR. The video and high-quality audio streams were synced using Adobe Premier Pro using the lower quality audio recorded from the video recorder to guide the audio alignment. Sentences with speech errors were eliminated, leaving 120 unique sentences. Participants were always presented with simultaneous audio-video stimuli.

### 2.2. Participants

A total of 83 listeners were recruited from undergraduate linguistics courses and received partial course credit in exchange for their participation. There were 66 female and 17 male participants between 18 and 26 years of age (Mean = 20). Listeners were either first language or early learners of English, which we operationalize as before the age of 5. Listeners self-reported their ethnicities (33 = Asian, 23 = White, 9 = South Asian, 3 = Indian, 2 = Asian and White, 1 = Asian Pacific Islander, 1 = Filipino, 1 = Japanese Canadian, 1 = Middle Eastern, 1 = First Nations, 1 = South East Asian and White; ethnicity information was missing for 7 participants).

### 2.3. Procedure

Participants were seated in front of a computer in sound-attenuated cubicles for the duration of the experiment. Listeners heard each sentence over headphones at approximately 65 dB while watching accompanying video of the talker on the screen. They were asked to type sentences on a keyboard and told to focus on being as accurate as possible while not worrying about minor spelling errors.

To facilitate adaptation to each individual talker, the task was blocked by talker. In each block, listeners heard 30 sentences from each talker. In order to control for talker order, there were 24 different permutations of the experiment. These orders were implemented cyclically, such that one participant would have order A and the next order B, resulting in approximately three to four participants for each. The 30 sentences were separated into 15 high predictability and 15 low predictability blocks, randomly selected for each listener. The high and low predictability blocks are thus designed and analyzed as training and test blocks, respectively. There were breaks between talkers, but not between sentence types within a talker.

This within-subject design for all talkers allows us to ignore talker-specific differences in intelligibility and focus on change—improvement or decline in performance—between high and low predictability blocks for each of the four talkers.

## 3. Results

### 3.1. Analyses

The measure of interest in this study is the change in listeners’ accuracy in transcribing a talker’s speech in noise between the set of high predictability sentences and the set of low predictability sentences. Transcription accuracy was automatically scored using the Token Sort Ratio, which is a fuzzy logic matching metric Bosker [32].

Data were analyzed with a Bayesian multilevel regression model using *brms* [33] in R using *cmdstanr* on the back end [34]. The model syntax was: TSR Sentence Predictability * Talker + (Sentence Predictability * Talker|Subject) + (1|Sentence), phi Sentence Predictability * Talker), family = Beta(). The TSR score is similar to proportion correct, and is bounded between 0 and 1, so a beta regression was used [35]. The actual 0 and 1 values were “squeezed” following the formula provided by Smithson and Verkuilen [35]. These squeezed TSR scores were the dependent measure. For the mean parameter, sentence predictability and Talker were the population-level (the “fixed effects” in frequentist mixed effects modeling jargon) effects, the interaction of which was also included in the model. Both sentence predictability and talker were dummy coded with high predictability sentences and the white English-L1 speaker as the reference levels. Listener and sentence were the group-level effects (i.e., the “random effects” in a frequentist model). There was a random intercept for sentence, while the listener-level effects included a random intercept and random slopes for sentence predictability, talker, and their interaction. Beta regression models include a phi parameter, which models the variance of the TSR scores. Sentence predictability and talker (without their interaction) were included as the population-effects for the phi parameter. Priors for all population-level effects were weakly informative priors of normal distributions with a mean of 0 and standard deviations of 2 and 1 for the intercept and population-level parameters, respectively, for both the mean and the phi components of the analysis. The standard deviations for the group-level effects had an exponential distribution of rate 1 as priors, and correlations used an LKJ prior of concentration 1. The model was fit using 4 Markov chains and 4000 samples each with 1000 warm-up samples per chain.

There were no divergent transitions and the $\hat{R}$ values were all <1.01, suggesting well-mixed chains. Inspection of the graphical posterior predictive check indicated that the model fit the data well.

Bayesian analysis allows for more nuance in evaluation evidence. When the 95% Credible Interval (CrI) for a given parameter excludes 0, this is considered strong evidence for an effect. The evidence for an effect is described as weak if the CrI includes 0, but the probability of direction is more than 95%. These choices follow Nicenboim and Vasishth [36].

### 3.2. Empirical Observations

The empirical data are presented as a box-and-whisker plot in Figure 1, with intelligibility—the Token Sort Ratio (TSR) score on the y-axis and the four talkers along the x-axis. Separate boxes visualize the distribution of responses for the high predictability training sentences (dark purple boxes) and the low predictability test sentences (yellow boxes). The empirical data suggest a generalization from the high to low predictability sentences for the Asian English-L2 and white English-L1 talkers, the talkers with stereotypically congruent race/accent associations. The median intelligibility score for these talkers is maintained across the two sentence types or, in the case of the Asian English-L2 increases across the testing and training, suggesting robust adaptation and generalization. The empirical data for the stereotypically incongruent Asian English-L1 and white English-L2 talkers shows a loss of intelligibility across the high predictablity training sentences and the low predictability test sentences, suggesting that listeners are less able to adapt to the noise for the talkers with stereotypically incongruent race/accent pairings.

### 3.3. Bayesian Regression

The empirical observations are quantitatively assessed through a Bayesian regression model. The $\hat{\beta }$ Estimate, standard error, 95% CrI, and the Probability of Direction of the fixed effects for this model are summarized in Table 2. The range of the posterior predictive distributions are plotted alongside the empirical data in Figure 1.

The high predictability utterances provided strong training for the low predictability utterances for the white English-L1 talker, who is the reference level in the model. There is no evidence for a loss of intelligibility across the training and test sentences [$\hat{\beta }=0,CrI=\left[-0.2,0.2\right],Pr(\hat{\beta }>0=0.52$]. The white English-L1 talker’s high predictability sentences were lower intelligibility than those of the Asian English-L1 talker [$\hat{\beta }\;=\;0.82,CrI=\left[0.71,0.94\right],Pr(\hat{\beta }>0=1$] and the white English-L2 talker [$\hat{\beta }\;=\;0.37,CrI=\left[0.26,0.48\right],Pr(\hat{\beta }>0=1$]. The white English-L1 talker’s high predictability sentences had higher intelligibility than the Asian English-L2 talker’s high predictability sentences [$\hat{\beta }=-0.57,CrI=\left[-0.69,-0.45\right],Pr(\hat{\beta }<0=1$]. Of primary interest, however, are the interactions between predictability and the talkers. While the empirical data and the point estimate suggest a loss of intelligibility for the Asian English-L1 talker across the test and training sentences compared to the white English-L1 talker, the CrI contains 0 and the probability of direction is relatively low [$\hat{\beta }=-0.07,CrI=\left[-0.2,0.05\right],Pr(\hat{\beta }<0=0.88$]. The results for the L2 speakers of English are in line with predictions, however. There is strong evidence that the high predictability sentences provided robust training for the stereotypically congruent Asian English-L2 talker [$\hat{\beta }=0.27,CrI=\left[0.14,0.41\right],Pr(\hat{\beta }>0=0.99$]. Likewise, the evidence is strong the the high predictability training sentences do not generalize to test performance with the low predictability sentences for the stereotypically incongruent white English-L2 talker [$\hat{\beta }=-0.29,CrI=\left[-0.42,-0.16\right],Pr(\hat{\beta }<0=1$].

Precision is inversely related to variance. Positive distributions for the phi parameter indicates an increase in precision, meaning that listeners were more consistent in the accuracy of their responses, while negative distributions indicate less precision and more variance. The positive distribution for the Low Predictability sentences with a CrI that does not contain 0 provides strong evidence that listeners were more consistent in the accuracy of their responses for the low predictability sentences compared to the high predictability sentences for the white English-L1 talker, who was the reference level [$\hat{\beta }=0.2,CrI=\left[0.14,0.27\right],Pr(\hat{\beta }>0=1$]. Recall the analysis of the mean indicated that the Asian English-L1 and the white English-L2 talkers’ high predictability sentences were overall more intelligible than the white English-L1 talker’s. For the Asian English-L1 talker, this is accompanied by strong evidence for high precision in her high predictability sentences [$\hat{\beta }=0.23,CrI=\left[0.13,0.32\right],Pr(\hat{\beta }>0=1$]. Listeners were more consistent in their accuracy to her high predictability utterances. There is weak evidence that listeners were more precise in transcribing the high intelligibility sentences for the white English-L2 talker compared to the reference level [$\hat{\beta }=0.08,CrI=\left[-0.01,0.16\right],Pr(\hat{\beta }>0=0.96$]. However, there was strong evidence that the Asian English-L2 talker, who had the least intelligible voice, elicited lower precision and, thus, more variance in response accuracy from listeners compared to the white English-L1 talker [$\hat{\beta }=-0.15,CrI=\left[-0.23,-0.07\right],Pr(\hat{\beta }<0=0.99$].

## 4. Discussion

In parts of North America, stereotypes about ethnicity and accent associations present a socio-linguistic landscape where white individuals are licensed to be native speakers of English, whereas non-white individuals are presumed to be second language speakers of English. In the local context, individuals of Asian descent may be stereotypically associated with non-English language, an expectation that is developed in infancy [23]. Any individual in the local context—infant or otherwise—has the opportunity for rich and diverse input. Nearly 50% of individuals in the Greater Vancouver Area identify as a visible minority [37], and nearly 45% of individuals report an “immigrant” language as their mother tongue, which includes all languages other than Aboriginal languages (First Nations languages, Inuktitut, and Métis), English, and French [38]. Certainly, it is not the case that all individuals who speak a Canadian-census-labelled “immigrant” language identify as visible minorities. However, the sheer amount of diversity presents listeners with a wide range of ethnicity and accent associations. Some local suburbs have historically coupled ethnicity and language associations that are particularly strong. And, indeed, previous work has shown that when local adults are cued to consider local stereotypes about particular suburbs that are strongly associated with Chinese Canadian culture, they find white Canadian individuals’ speech more intelligible and less accented than the speech from Chinese Canadians *only* when they are aware they are listening to individuals who self-identify as Chinese Canadian [19]. In this previous work, individuals with stronger Asian Canadian social networks showed these effects more strongly, suggesting that negative bias towards Asian Canadians is unlikely to underlie the drop of intelligibility and increase in perceived accentedness. These results, along with others’ findings in the literature on ethnicity and accent associations (e.g., [18,24,29]), suggest that experiences may undergird ethnicity and accent associations more than negative social attitudes [16]. This scholarship led to the hypothesis that listeners should more readily adapt to degraded audio-visual speech when the talker’s ethnicity and accent align with local stereotypes. Specifically, this hypothesis offers the prediction that listeners will more readily adapt to a white L1-English speaker and an Asian L2-English (Mandarin-accented, in this case) speaker. The white L2-English speaker contradicts local expectations about white speakers, and listeners were predicted to struggle in their adaptation to her speech. While a diverse population is expected to speak the local mainstream variety of English [23], listeners may also implicitly carry the expectation that Asian individuals speech English with a non-native accent, which would entail listeners not adapting to the speech of the Asian L1-English speaker (e.g., [15,16,18,19]).

The current experiment tested this hypothesis space in an experiment that compared the change in intelligibility—measured by listeners’ accuracy in transcribing audio-visual speech in noise—between high predictability training sentences and low predictability test sentences. The data were analyzed using a Bayesian mixed effects beta regression model, which allows nuance in interpretation and a joint consideration of estimates for means and variance (the phi parameter). The four talkers varied in their respective baseline levels of intelligibility in the high predictability test sentences. The white English-L1 speaker had relatively low baseline intelligibility, and the Asian English-L1 speaker and the white English-L2 speaker were both more intelligible than the white English-L1 speaker in the high predictability test sentences. Anecdotally, the white English-L1 speaker’s speech style exhibited a relatively small amount of head and jaw motion, the movement of which is known to be beneficial for intelligibility [39]. The, perhaps, surprisingly low intelligibility of the white English-L1 speaker underscores the importance of using a paradigm that allows each individual and their own speech idiosyncrasies as their own control. The Asian English-L2 speaker was less intelligible than the white English-L1 speaker for the high predictability sentences.

It is the interactions between the sentence type and talker, however, that provide insight to the research question: is adaptation affected by accent and ethnicity associations? Starting with the clear results, listeners showed the predicted adaptation to the Asian English-L2 speaker, generalizing their experiences in the high predictability training sentences to the more challenging low predictability test sentences. This is in line with the stereotyped expectation that Asian individuals will have a non-native accent; with this expectation, listeners were able to leverage the expected non-native accent to more robustly learn and generalize from the high predictability sentences. Also in accordance with the prediction that listeners anticipate a white individual to speak English as a first language, listeners’ exhibited a loss of intelligibility across training and test sentences for the white English-L2 speaker. This indicates that despite the white English-L2 speaker being a clear talker, as evidenced by the high intelligibility in the high predictability test sentences, listeners were not able to adapt to her speech patterns. There is little-to-no evidence that listeners are unable to adapt to the Asian English-L1 speaker. While the mean point estimate is negative, suggesting a tendency towards a challenge to adapt, which would suggest an expectation of an Asian individual being a non-native English speaker, the 95% credible interval crosses 0 and the probability of direction is 88%. The effect size would be small and the spread of the 95% credible interval is wide, though not as wide as it is for the interaction between sentence type and the white English-L1 talker and the Asian English-L2 talker. This lack of an effect at the population-level suggests that listeners are prepared for an Asian individual to speak the local accent, an expectation that is well-suited to the multi-ethnic landscape of their speech community. These results are in line with infants’ developing expectations about the local native accent within this particular speech community [23], in addition to listeners’ flexibility with “standard” German accents in the German-speaking context [24].

As noted, the spread of the credible interval for the English-L2 talkers is wide, indicating a large amount of variation in response to these non-native accents. Indeed, as the listeners were sampled from the same speech community as the speakers, all parties are exposed to a wide variety of first and second (and beyond) language accents in the local university setting and metropolitan area. Individuals who have more direct or regular experience with Mandarin-accented and Spanish-accented Englishes, the varieties spoken by the Asian and white talkers in this study, would have representations that are better equipped to parse these talkers. Note, however, that despite experience, the expectations that faces and voices presented in a university laboratory would conform to stereotypes about ethnicity and accents were generally maintained.

A beta regression model requires the modeling of precision, which is inversely related to variance. This phi parameter suggested that listeners were more consistent, showing less variance, in their transcriptions of the Asian English-L1 talker, who also had higher baseline intelligibility than the white English-L1 talker. These results simply suggest that an English-L1 speaker with clear speech patterns is more uniformly intelligible to listeners. Speaking to the wide credible interval in the estimate for the means, the variance associated with transcription accuracy for the Asian English-L2 speaker was higher compared to the white English-L1 talker (the reference level for the statistical model). Listeners likely have more varied levels of experience with Mandarin-accented English, and this is evidenced in the range of credible intervals and the lower precision values. The evidence for less variance for the white English-L2 talker compared to the white English-L1 reference level was weak, which may be due to her higher baseline intelligibility coupled with her non-native accent, to which listeners will have varied experience.

While the age distribution of the listeners in the current study is quite constrained (18–26 years of age), the self-identified ethnicity of the participants is varied. This was not the result of targeted recruitment, but a representation of the local university setting and speech community in which this study was conducted. This participant diversity aligns with other studies on the role of stereotypes on speech intelligibility at the same university [19]. Note that other scholars who report listener self-reported ethnicity generally have a much less diverse sample in terms of ethnicity (e.g., [24,25,27]). With respect to age, however, Hanulíková et al. [40] used aged-diverse listeners (teens, young adults, and older adults) and found that accent-ethnicity stereotypes were more pronounced in older listeners. Given this, it may be that older listeners in the local speech community would exhibit stronger effects than the young adult population used in the current study. Such a prediction, however, would depend on older listeners’ having the requisite experiences to generate such predictions about accent and ethnicity associations.

The current results provide empirical behavioural support for a model of speech processing where listeners buffer their linguistic expectations about the incoming speech stream based on socio-cultural information about the speaker. Support for the notion that listeners have expectations about speakers also exists, however, in neurolinguistic measures. Listeners make socio-demographic assessments about an individual based on their voice (e.g., deducing from the speech signal that a talker is an adult or child) and parse the pragmatic appropriateness of their utterances (e.g., a child’s or adult’s voice saying “Every evening I drink some wine before I go to sleep.”). Van Berkum et al. [41] found an N400 effect when the speaker and their linguistic message were pragmatically inconsistent. Listeners also have knowledge and expectations about linguistic forms as they relate to an individual’s accent. Hanulíková et al. [40], for example, found that L1 Dutch listeners exhibited a P600 effect in response to a grammatical error when the voice producing the error was a Dutch-L1 accent, but not when the voice had a Turkish-accented Dutch accent. They reason that since the grammatical error is typical for Turkish-L1 Dutch-L2 speakers, listeners anticipated the grammatical error, hence the lack of a P600. Recently, Zhou et al. [42] demonstrated that the expectations about the grammatical patterns of natively accented and non-natively accented from imagined speech — i.e., auditory perceptual simulation. (Zhou et al. [42] presented participants with a photo of a white female and an Asian female, and they report presenting these photos with recordings from “native” and “non-native” female speakers. The voices and photos were further accompanied by an English name and a Chinese name, so presumably the “non-native” accent was Mandarin-accented English.) Listeners exhibited a P600 effect when imagining the L1 English voice producing a sentence with a grammatical error, but showed no P600 effect when the same utterance was imagined in a Mandarin-accented English voice.

Given the local linguistic diversity, it is assumed that for the current population the stereotypes about accent and ethnicity are at least partially formed by experience and not media-mediated linguistic stereotypes, though such input may indeed play a critical role in seeding associations. The observed ethnicity-accent associations are certainly over-generalizations in the sense that they are often erroneous. Melguy and Johnson [27] discuss, for example, how a generic association of white talkers with native English accents is incompatible in English-speaking communities with a large number of white immigrants from non-English-speaking countries (e.g., their example is immigration in the dissolution of the Soviet Union). We can add to this, of course, that in many countries, particularly those with long-standing colonial histories or those with large amounts of immigration, non-white individuals are native speakers of a majority language like English. While expectations may be part and parcel of the organization of the linguistic processing system, we need awareness of these affects to mitigate their social costs. It is simply unfair for linguistic stereotyping to impose an intelligibility cost on particular ethnicity-accent associations.

Lastly, while audio-visual methods are far from novel, their application in social linguistic inquiries is, arguably, under-utilized. The use of audio-visual speech to probe the role of previous experience and the use of phonetic and phonological expectations in the processing of speech offers the distinct advantage of being more ecologically valid. Behaviours gathered in the context of audio-visual speech, as opposed to static photos, may also be less susceptible to strategies and task effects [21].

## 5. Conclusions

In a multicultural and multilingual speech community, listeners exhibit accent and ethnicity associations. The local variety of English is spoken by a diverse demographic, and listeners are flexible with English-L1 accents, whether spoken by white or Asian English-L1 speakers. For non-native accents, listeners adapted to the accent and ethnicity association that conforms to local stereotypes (an Asian English-L2), but not to an incongruent association (a white English-L2). These results provide support for accounts where intelligibility is supported by the alignment of phonetic and phonological expectations with the apprehended phonetic signal.

## Figures and Tables

**Figure 1 brainsci-12-00845-f001:**
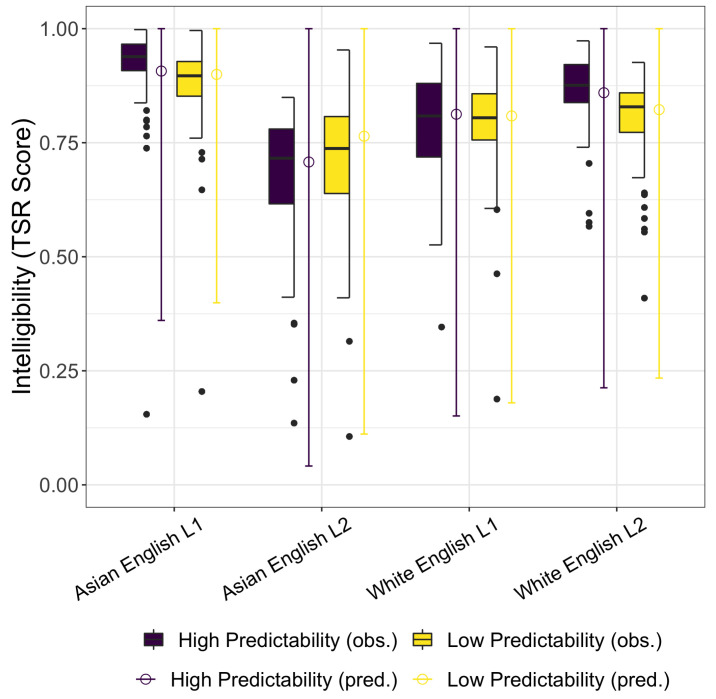
A box-and-whisker plot of the empirical results and the range of the posterior predictive distributions for intelligibility, plotted as TSR scores, for the four talkers separated by high and low predictability sentences.

**Table 1 brainsci-12-00845-t001:** Example high and low predictability sentence stimuli.

Sentence	Predictability
The opposite of hot is cold.	High
For your birthday, I baked a cake.	High
In the spring plants are full of green leaves.	High
He pointed at his hair.	Low
Mom thinks that is yellow.	Low
She talked about the leaves.	Low

**Table 2 brainsci-12-00845-t002:** Population-level or fixed-effect predictors for the beta regression model. The $\hat{\beta }$ estimate, standard error, and 95% Credible Interval (CrI) for TSR are reported for the means and phi parameters.

*Mean*				
	$\hat{\beta }$ Estimate	Standard Error	95% CrI	Probability of Direction
Intercept	1.4553	0.0857	[1.29, 1.62]	1
Low Predictability	−0.0030	0.1013	[−0.2, 0.2]	0.52
Asian English-L1	0.8227	0.0619	[0.71, 0.94]	1
Asian English-L2	−0.5695	0.0612	[−0.69, −0.45]	1
white English-L2	0.3731	0.0594	[0.26, 0.48]	1
Low:Asian English-L1	−0.0734	0.0633	[−0.2, 0.05]	0.88
Low:Asian English-L2	0.2717	0.0690	[0.14, 0.41]	0.99
Low:white English-L2	−0.2901	0.0594	[−0.42, −0.16]	1
* **phi** *				
	$\hat{\beta }$ Estimate	Standard Error	95% CrI	Probability of Direction
Intercept	0.4101	0.0340	[0.34, 0.48]	1
Low Predictability	0.2043	0.0318	[0.14, 0.27]	1
Asian English-L1	0.2289	0.0490	[0.13, 0.32]	1
Asian English-L2	−0.1530	0.0410	[−0.23, −0.07]	0.99
white English-L2	0.0768	0.0444	[−0.01, 0.16]	0.96

## Data Availability

Data and code are available at https://osf.io/pwj6b/?view_only=290b33be56f648eaa26f51fea27e5ee2 (accessed on 24 May 2022).

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
