# Peer review of "Adaptation to Social-Linguistic Associations in Audio-Visual Speech"

_brainsci, 2022, doi:10.3390/brainsci12070845_

Round 1
Reviewer 1 Report
The study mainly reproduces results from earlier studies, but with an adjusted methodology. In the conclusion, it might be good to include a bit more about how these results are distinct from those of prior researchers. As it stands, it downplays prior findings rather than integrating these results into the literature. Other wise, my main suggestion would be that the author discuss the role of age in the experiment. Earlier in the paper, a study is referenced showing that the "rubin effect" is stronger for older speakers. The study has all younger speakers which fits with the earlier study. Some sort of caveat or discussion of the motivation for using younger speakers would be useful. Also, a bit more discussion of the motivations for placing speakers into blocks (rather than having them randomized) would be beneficial.
Author Response
The study mainly reproduces results from earlier studies, but with an adjusted methodology. In the conclusion, it might be good to include a bit more about how these results are distinct from those of prior researchers. As it stands, it downplays prior findings rather than integrating these results into the literature. Other wise, my main suggestion would be that the author discuss the role of age in the experiment. Earlier in the paper, a study is referenced showing that the "rubin effect" is stronger for older speakers. The study has all younger speakers which fits with the earlier study. Some sort of caveat or discussion of the motivation for using younger speakers would be useful. Also, a bit more discussion of the motivations for placing speakers into blocks (rather than having them randomized) would be beneficial.
> I break down each comment below for a response.
In the conclusion, it might be good to include a bit more about how these results are distinct from those of prior researchers. As it stands, it downplays prior findings rather than integrating these results into the literature.
> I have reworked the discussion to better integrate the current results within the prior literature.
The study has all younger speakers which fits with the earlier study. Some sort of caveat or discussion of the motivation for using younger speakers would be useful.
> The use of younger speakers was simply a convenience sample from our university subject pool. I discuss the age of listeners more directly in the discussion.
Also, a bit more discussion of the motivations for placing speakers into blocks (rather than having them randomized) would be beneficial.
> The decision to block by talker was made to facilitate learning for each talker. A full randomization would have been excessively difficult, given the adverse noise conditions.
Reviewer 2 Report
The work examines social-linguistic associations in audio-visual speech. This is a productive and developing area of research and the manuscript contributes to the new knowledge in the field by elaborating the speakers’ profiles and extending the previous findings to different communities.
Major points:
1. Lit Review:
a. You need to add another reference in the lit review: Zhou, P., Garnsey, S., & Christianson, K. (2019). Is imagining a voice like listening to it? Evidence from ERPs. Cognition, 182, 227-241. They have the most direct evidence that speakers adjust their expectations regarding speech depending on their native-ness perception for the speaker. This work showed that when comprehenders knew that the speaker is English L2 they did not treat the grammatical mistakes as mistakes – i.e. no P600 in response to an English ungrammatical sentence. However, when they knew that the speaker was a native English speaker the authors observed a typical P600 in response to an ungrammatical sentence.
2. Materials:
a. Please add examples of high and low predictability sentences. This is one of the central manipulations of the manuscript. It is not enough to add the citation and make your readers dig for the examples in other publications.
3. Participants:
a. Please add the race distribution across participants
4. Results:
a. Please add Bayes Factors H0 & H1 for Table 1
b. Why white English L1 talker’s high predictability sentences had lower intelligibility than those of Asian English L1 talker need to be explained more clearly. Intuitively, readers would expect the results to be reversed.
Minor:
P2. Line 88 – remove second word “talker”
P4. Line 184 – change babel2015 to “Babel (2015)”
Author Response
Thank you for these constructive comments.
Major points:
- Lit Review:
- You need to add another reference in the lit review: Zhou, P., Garnsey, S., & Christianson, K. (2019). Is imagining a voice like listening to it? Evidence from ERPs. Cognition, 182, 227-241. They have the most direct evidence that speakers adjust their expectations regarding speech depending on their native-ness perception for the speaker. This work showed that when comprehenders knew that the speaker is English L2 they did not treat the grammatical mistakes as mistakes – i.e. no P600 in response to an English ungrammatical sentence. However, when they knew that the speaker was a native English speaker the authors observed a typical P600 in response to an ungrammatical sentence.
> Thank you for bringing this paper to my attention. I now connect the current results to the Zhou et al. paper and the body of work it falls into.
- Materials:
- Please add examples of high and low predictability sentences. This is one of the central manipulations of the manuscript. It is not enough to add the citation and make your readers dig for the examples in other publications.
> I should have included these in the first place. I now include example sentences in Table 1, and provide a link to the full list in the OSF directory associated with this project.
- Participants:
- Please add the race distribution across participants
> Ethnicity information for the participants has now been added. The diversity of the participant population is now also addressed in the discussion.
- Results:
- Please add Bayes Factors H0 & H1 for Table 1
> Bayes Factors are very sensitive to the selected priors, which removes the ability to weigh the evidence supporting a hypothesis in a more nuanced way, as one can when the Credible Interval and Probability of Direction are jointly interpreted. Given this, I have opted not to add Bayes Factors to the analysis. In revisiting Table 1 (now Table 2), however, I did notice that failed to include the standard error from the model. Those values are now included.
- Why white English L1 talker’s high predictability sentences had lower intelligibility than those of Asian English L1 talker need to be explained more clearly. Intuitively, readers would expect the results to be reversed.
> This is an important point. Indeed, some might assume that the white English-L1 speaker would be the most intelligible, but different talkers vary, of course, in their intelligibility in ways that are unrelated to to their socio-demographic characteristics. I now offer my speculation about this particular talker’s comparatively low intelligibility (she is a kind of “closed mouth” talker), and provide a reference for the role of head and jaw movement in boosting intelligibility.
Minor:
P2. Line 88 – remove second word “talker”
> Fixed.
P4. Line 184 – change babel2015 to “Babel (2015)”
> Fixed.